# Antimicrobial Impacts of Microbial Metabolites on the Preservation of Fish and Fishery Products: A Review with Current Knowledge

**DOI:** 10.3390/microorganisms10040773

**Published:** 2022-04-03

**Authors:** Nikheel Bhojraj Rathod, Nilesh Prakash Nirmal, Asif Pagarkar, Fatih Özogul, João Miguel Rocha

**Affiliations:** 1Department of Post Harvest Management of Meat, Poultry and Fish, PG Institute of Postharvest Management, Dr. Balasaheb Sawant Konkan Krishi Vidyapeeth, Roha, Raigad 402116, Maharashtra, India; nikheelrathod310587@gmail.com; 2Institute of Nutrition, Mahidol University, 999 Phutthamonthon 4 Road, Salaya, Nakhon Pathom 73170, Thailand; nilesh.nir@mahidol.ac.th; 3Marine Biological Research Station, Dr. Balasaheb Sawant Konkan Krishi Vidyapeeth, Ratnagiri 415612, Maharashtra, India; pagarkarau@gmail.com; 4Department of Seafood Processing Technology, Faculty of Fisheries, Cukurova University, Adana 01330, Turkey; 5LEPABE—Laboratory for Process Engineering, Environment, Biotechnology and Energy, Faculty of Engineering, University of Porto, Rua Dr. Roberto Frias, 4200-465 Porto, Portugal; 6ALiCE—Associate Laboratory in Chemical Engineering, Faculty of Engineering, University of Porto, Rua Dr. Roberto Frias, 4200-465 Porto, Portugal

**Keywords:** antimicrobials, metabolites, biopreservation, seafood, food security, foodborne pathogens, bacteriocins, organic acids, reuterins, bacteriophages and endolysins, high pressure processing, modified atmosphere packaging

## Abstract

Microbial metabolites have proven effects to inhibit food spoilage microbiota, without any development of antimicrobial resistance. This review provides a recent literature update on the preservative action of metabolites derived from microorganisms on seafood. Fish and fishery products are regarded as a myriad of nutrition, while being highly prone to spoilage. Several proven controversies (antimicrobial resistance and health issues) related to the use of synthetic preservatives have caused an imminent problem. The demand for minimally processed and naturally preserved clean-label fish and fishery products is on rise. Metabolites derived from microorganisms have exhibited diverse preservation capacities on fish and fishery products’ spoilage. Inclusions with other preservation techniques, such as hurdle technology, for the shelf-life extension of fish and fishery products are also summarized.

## 1. Introduction

The application of microorganisms and their metabolites for the preservation of foods is known as biopreservation [1,2,3,4]. Biopreservation is an emerging field, having applications along the whole food sector for the processing and preservation of foods. Food spoilage is a global issue associated with impacts on the supply of food, nutrition and impacting human health. Microbial spoilage of food has several associated microorganisms that could cause food poisoning. According to the World Health Organization (WHO) food-borne illness accounts for around 600 million infections and 420 thousand deaths per year. The majority of contributors are pathogenic *Vibrio*, *Listeria monocytogenes*, *Clostridium botulinum* and histaminogenic bacteria, mainly found on spoiled FFPs. Synthetic preservatives (triclosan, butylated hydroxyanisole, butylated hydroxytoluene, parabens, and sulfites) employed have been reported to induce resistance, toxicity and health disorders [5,6,7,8,9,10,11,12,13]. Considering these adversities, consumer trends have been shifted towards foods preserved using naturally occurring compounds individually or in combination with other novel processing methods for inactivating foodborne spoilage and pathogenic microorganisms [14,15]. This has diverted researcher attention towards the biopreservation of the seafood products. However, to be qualified as a biopreservative, microorganisms or their metabolites should have a wide antimicrobial effect against spoilage causing and pathogenic microorganisms, as well as be safe for the consumers. Additionally, biopreservative microorganisms and/or their metabolites should be stable at various processing conditions and should not alter the sensory properties of the food [16].

In particular, lactic acid bacteria (LAB) have widely been explored for their antimicrobial characteristics related to the release of secondary metabolites and compounds produced, exhibiting a preservative effect [17,18,19,20,21,22,23,24,25,26]. Bacteriocins are antimicrobial peptides produced outside the cells by microorganisms (Gram-positive and Gram-negative), restricting the growth of undesirable microorganisms, e.g., nisin, pediocin and reuterin [22]. Apart from antimicrobial effects, their ability to enhance quality when applied has also been reported [27,28]. Organic acids are products of microbial metabolism, exhibiting antimicrobial and antioxidative properties and having a generally recognized as safe status (GRAS), bearing one or more carboxylic groups in their molecule [29,30,31,32]. Recently, the United States Department of Agriculture’s (USDA) food safety inspection services has approved the application of organic acids (5%) followed by a water rinse in a pre-eviscerated state and final levels not exceeding specified levels (0.25%—sodium diacetate, 5%—lactic acid) in finished products [33,34]. Metabolites originated from the lactic acid bacteria genera are mostly used for the preservation of seafood [17] due to their GRAS (generally recognized as safe) status and no adverse issues reported. Considering regulatory approval, bacteriocin and nisin were the microbial metabolites approved for food preservation by the USA Food and Drug Administration in 1988 [35] and the European Union in 2011 [36], respectively. The application of Bacteriophage Listex P100 has been approved by the USA Food and Drug Administration as additive in food intended for human consumption [37]. A slight paucity in the literature is found due to their impacts on the sensory qualities of fish and fishery products (FFPs), however, the negative impacts on sensory qualities could be resolved by the inclusion of another appropriate processing system, such as encapsulation [38,39].

Considering the increased production and *per capita* consumption (9.9 to 20 kg) of FFPs during the last six decades, it exhibits further potential as per the suggestion of global health agencies [40,41,42]. These higher consumption rates are linked to their higher nutritional qualities attributed to the presence of quality fats (5–20%), rich in unsaturated fatty acids (200–500 mg of mega-3 polyunsaturated fatty acids from 1 to 2 servings), proteins (15–20%), rich in essential amino acids and highly absorbable in the human body, and the presence of vitamins (A, B complex and D) and minerals (calcium, selenium, zinc, iron, phosphorus, potassium and iodine) [43,44,45]. However, the shorter shelf-life (3–5 days) of FFPs is a matter of concern due to cellular breakdown by endogenous and exogenous (phosphorylase, lipases, phospholipases, lipoxygenases, hydroperoxidases, proteases) enzymes, and also, microorganisms from both the Gram-positive and -negative class (*Shewanella*, *Pseudomonas*, *Photobacterium*, *Aeromonas*, *Bacillus*, *Brochothrix*, *Carnobacterium*, *Escherichia*, *Enterobacter*, *Listeria*, *Micrococcus*, *Moraxella*, *Proteus*, *Psychrobacter*, and *Vibrio*) cause degradation [15,43,46,47]. The preservation of FFPs has gained immense importance due to an increased demand in the market globally. Hence, antimicrobials derived from synthetic sources have been used to serve the purpose. Due to their discussed limitations and unregulated usage, as discussed in our earlier review [15], their application is nearing stagnation. Furthermore, the unregulated usage of synthetic preservatives has led to the development of resistance amongst the microbial population and the development of several disorders in humans as well—which has led to consumer orientation shifting towards FFPs processed minimally and reserved using naturally derived preservatives [7,9,11,12]. The consumer orientation towards foods preserved by natural ingredients is regarded as healthy and is valued by consumers. Additionally, several novel non-thermal techniques, such as cold plasma, a pulsed electric field and high hydrostatic pressure, are widely under evaluation for the preservation of FFPs individually or in combination with natural preservatives [47,48].

Considering the consumer orientation towards FFPs preserved using natural preservatives, metabolites produced by microorganisms such as bacteriocins and organic acids for their protection purpose could be good alternatives to synthetic preservatives. Due to their ability to inhibit a wide range of spoilage and pathogenic microorganisms, their non-toxic nature, heat stable nature and having no detrimental impacts on food products, metabolites from a microbial origin are gaining great importance. In this regard, biopreservation has gained momentum for application in FFPs for their preservation. Therefore, this review provides up-to-date information on the antimicrobial activity of microbial metabolites, such as bacteriocins and organic acids, for the preservation of FFPs.

## 2. Microbial Spoilage of Fish and Fishery Products

FFPs contain a wide array of microorganisms from different environments. These microorganisms lead to the spoilage of FFPs and pose health risks to consumers. Fish surfaces contain the microbiota, including natural microorganisms of the waters, from which fish are harvested, as well as acquired cross contamination microbiota which includes microorganisms entering the food from fish’s contact surfaces, the air, soil, water/ice used for washing/fish handlers, packaging material and the storage environment [15,17,49]. Generally, microorganisms are mainly associated with the outer slime, gills and intestines [49]. The microbial load is higher in the intestines, followed by the gills and skin (102–107 CFU/cm^2^). Normally, the microbial load ranges from 10^2^–10^7^ colony-forming units (CFU)/cm^2^ on the skin’s surface [15], whereas in the gills and the intestines, both contain between 10^3^ and 10^9^ CFU/g [50,51].

Fresh healthy fishes are usually sterile as the immune system of the fish prevents bacteria from growing in the flesh. After the death/harvest of fish during storage, the microorganisms invade the flesh by moving between the muscle fibers. The natural microbiota of fish varies depending on the habitat of the fish, whether freshwater, marine or brackish water, its feeding habit and its life history stages. Generally, warm water fish have more mesophilic bacteria than cold water fish. Fish harvested from polluted/contaminated water contain a variety of microorganisms depending on the nature of the pollutant/contaminant, and also human pathogens such as bacteria, fungi, viruses, protozoans, parasites, etc. [52]. Very high microbial numbers of 10^7^ CFU/cm^2^ are found on fish from polluted warm waters. They play an important role in spoilage and lead to food poisoning. Thus, it is necessary to maintain the quality of fish and fish products by inhibiting the associated microorganisms and preventing their growth. The different preservation methods mainly aim at maintaining fish quality by reducing, killing or inactivating associated spoilage microorganisms [15].

In general, Gram-negative, psychrotrophic, rod-shaped bacteria belonging to the genera *Pseudomonas*, *Moraxella*, *Acinetobacter*, *Shewanella* and *Flavobacterium* are dominated in temperate water fish, whereas members of the Vibrionaceae (*Vibrio* and *Photobacterium* spp.) and the Aeromonadaceae (*Aeromonas* spp.) are common of the fish microbiota [47,48]. Gram-positive bacteria, such as *Bacillus*, *Micrococcus*, *Clostridium*, *Lactobacillus* and coryneforms, can also be found in varying proportions, but in general, Gram-negative bacteria dominate the microbiota [53,54]. Gram-positive *Bacillus* and *Micrococcus* were found to dominate the fish from tropical waters [55,56]. Microbiota consisting of *Pseudomonas*, *Acinetobacter*, *Moraxella* and *Vibrio* spp. have been observed on newly-caught fishes [57,58]. In polluted waters, high numbers of Enterobacteriaceae may be found. In clean temperate waters, these microorganisms disappear rapidly, but it has been shown that *Escherichia coli* and *Salmonella* spp. can survive for very long periods in tropical waters and once introduced, may almost become indigenous to the environment [59].

The growth of microorganisms and their metabolism is a major cause of fish spoilage as they produce biogenic amines such as putrescine, histamine and cadaverine, organic acids, sulfides, alcohols, aldehydes and ketones with unpleasant and unacceptable off-flavors [60,61,62]. Spoilage is a combined result of Gram-negative fermentative bacteria (*Vibrionaceae*). Gram-negative, psychrotolerant bacteria (*Pseudomonas* and *Shewanella* spp.) tend to spoil unpreserved or chilled fish [15,63].

Trimethylamine (TMA) levels are used universally to correlate with the extent of microbial deterioration responsible for fish spoilage [15,64]. Trimethylamine oxide (TMAO) is used as an osmoregulant to avoid dehydration in marine environments [15]. Bacteria such as *Shewanella putrifaciens*, *Aeromonas* spp., psychrotolerant Enterobacteriacceae, *Pseudomonas phosphoreum* and *Vibrio* spp. can obtain energy by reducing TMAO to TMA, creating the ammonia-like off-flavors [62], as shown in Figure 1. *Pseudomonas putrefaciens*, fluorescent pseudomonads and other spoilage bacteria increase rapidly during the initial stages of spoilage, producing many proteolytic and hydrolytic enzymes [65,66].

The total volatile base (TVB-N) rises even after TMA has reached its maximum, which is due to proteolysis, and it starts when several of the free amino acids have been used. Lerke et al. [67] separated fish juice into a protein and a non-protein fraction and inoculated spoilage bacteria in both fractions. The non-protein fraction of fish juice spoiled, as did the whole juice, whereas only faint off-odors were detected in the protein fraction of the juice. Bacteria action during the spoilage of fish used the substrate for the formation of compounds, i.e., from substrate amino acids (glycine, serine, leucine), several esters, ketones, aldehydes, amino acids, urea, cysteine, carbohydrates and lactate acetate, carbon dioxide, water, TMAO, TMA, and methionine were produced by bacteria.

The anaerobic storage of fish for a long-time results in vigorous the production of ammonia owing to the degradation of the amino acids and the accumulation of lower fatty acids (acetic, butyric and propionic acids) [15,17]. Obligate anaerobes belonging to the family Bacteroidaceae and the genus *Fusobacterium* were found to be very strong ammonia producers [68]. These bacteria grew only in the spoiled fish extract and have little or no proteolytic activity relying on already hydrolyzed proteins.

## 3. Antimicrobial Mechanism of Microbial Metabolites

### 3.1. Bacteriocins

Bacteriocins are ribosomal synthesized proteins or peptides with a bactericidal or bacteriostatic action [69]. Due to their antimicrobial activity against several spoilage and pathogenic microorganisms, bacteriocins are widely under evaluation for the preservation and shelf-life extension of foods [70,71]. Both Gram-positive and Gram-negative bacteria are known to produce bacteriocins. Bacteriocins exhibit a diverse preservative action by diverse mechanisms, such as their positive charge which interacts with negatively charged cell wall constituents forming a pore in the cell wall, and inhibit the synthesis of the cell wall [72,73]. In some cases, the terminal end of the peptide group of bacteriocins causes leaking of cellular constituents, and inhibits protein synthesis and DNA replication [74,75]. Bacteriocins have recently been reported to hamper the replication of bacteria cells by inhibiting the formation of a septum in the bacteria [72,76]. Apart from this, they are also known to compete for nutrients and produce secondary metabolites which further extend the inactivation of microorganisms [77]. Bacteriocins have been classified into three classes based on their molecular weight and post-translational modification, imparting on them different antimicrobial actions [31,78,79]. Class-I bacteriocins are post-translationally modified heat-stable peptides with a molecular weight of less than 10 kDa. Further, Class-I bacteriocins are subdivided into six subclasses of Ia to If [80]. Class-II bacteriocins consist of unmodified heat-stable peptides with a molecular weight of less than 10 kDa and subdivided into Class-IIa to IId types. Class-III bacteriocins are unmodified heat-sensitive proteins with a molecular weight of more than 10 kDa and include two subtypes (IIIa and IIIb) [80]. Several techniques, including bacteriocins for the preservation of FFPs, have been given in Table 1. Recently, the trend to extract bacteriocins from seafood and their possible effects in preserving seafood has been gaining importance [81]. However, very few microbial metabolites, including nisin, pediocins, lacticin, reuterin, and organic acids, have been studied as natural preservatives for seafood products. Among the studied bacteriocins, so far, only nisin has been approved and accepted as a food additive by the European Food Regulation Committee and other countries [30,82].

Nisin is classified as a Class-Ia (lantibiotic) and produced by the *Lactococcus lactis* subsp. *lactis*. Nisin is composed of 34 amino acid residues with a molecular weight of 3.5 kDa [80]. Recently, nisin has been investigated in combination with other hurdles to enhance the biopreservative ability and stability of the nisin [30]. In their study, Chatzidaki et al. [83] reported that nisin-loaded nanocarriers showed enhanced antibacterial activity of *Lactococcus lactis*, *Staphylococcus aureus*, *Listeria monocytogenes* and *Bacillus cereus* by microemulsion formulation using essential oils as nano-carriers.

A combination of glazing incorporated with nisin and irradiation treatments at 2 kGy and 5 kGy were investigated for the shelf-life extension of seer fish (*Scomberomorous guttatus*) steaks [84]. The results showed that an increase in the irradiation dose increased the shelf-life of steaks from 7 days to 34 (2 kGy) and 42 (5 kGy) days. The effects of nisin in combination with high-pressure processing at a low temperature were evaluated against *Listeria innocua* survival and the quality attributes of dry-cured cold-smoked salmon (*Oncorhynchus nerka*) fillets during 36 days of refrigerated storage [85]. The nisin (10 µg/g) and high-pressure processing (HPP) at 450 and 600 MPa for 2 min effectively reduced the *L. innocua* growth. The application of the 450 MPa and 600 MPa pressure for 120 s had impacts on the quality of the dry-cured cold-smoked salmon by reducing *L. innocua* and improved sensory properties. Additionally, the lower temperature slightly increased the lightness of the cold-smoked salmon compared to the ambient temperature [85]. The treatment of tuna (*Thunnus albacares*) fillets with an osmotic solution containing nisin significantly improved the shelf-life of the fillets from 10 days to 51 days at 5 °C by lowering the microbial changes and improving the sensory properties during storage [86]. Nisin-loaded chitosan microcapsules were formulated and examined for their preservative ability of the small yellow croaker (*Pseudosciaena polyactis*) during refrigerated storage for 30 days [87]. The prepared microcapsules significantly extended the shelf-life of croaker by 6–9 days, by reducing the microbial growth, lipid oxidation and protein degradation, when compared to chitosan, nisin alone or the control. Yang, Cheng, Tong, and Chen [88] investigated the effect of nisin (0.1% solution) on the quality of tortoise meat (*Trachemys scripta elegans*) during chilled storage for 15 days. The results indicated that the nisin treatment significantly lowered the microbial and chemical changes while improving the sensory properties of the meat, thereby extending the shelf-life of the meat by 3–6 days. The effect of chitosan in combination with different concentrations of nisin was investigated for the quality control of a Large yellow croaker (*Pseudosciaena crocea*) during refrigerated storage of 8 days [89]. The results showed that chitosan (1%) with nisin (0.6%) showed a higher efficiency in the quality improvement of fish by lowering water loss, chemical and microbial changes, as well as maintaining the sensory attributes of the fish. Woraprayote et al. [90] synthesized a biodegradable packaging material (poly lactic acid/sawdust particles) incorporated with bacteriocin 7293 isolated from *Weissella hellenica* BCC 7293. Further, the antimicrobial properties of packaging film were determined using an in vitro assay in pangasius fish fillets. The antimicrobial film inhibited the growth of Gram-positive (*L. monocytogens*—4.78 log CFU/cm^2^ and *S. aureus*—4.56 log CFU/cm^2^) and Gram-negative (*Pseudomonas aeruginosa*—4.10 log CFU/cm^2^, *Aeromonas hydrophila*—3.12 log CFU/cm^2^, *Escherichia coli*—3.03 log CFU/cm^2^ and *Salmonella thyphimurium*—3.03 log CFU/cm^2^) bacteria in both the in vitro assay and a challenge test with fish fillets [90].

Similarly, lacticin, a two peptide bacteriocin, was reported to be produced by the *Lactococcus lactis* subsp. *lactis*. Kim et al. [91] showed the ability of lacticin NK24 against spoilage and pathogenic bacteria isolated from jeotgal, a Korean fermented fish food. Effective inhibitions against Gram-positive bacteria were observed in comparison to Gram-negative bacteria [91]. Similarly, a significant inhibition of total aerobic bacteria and coliform bacteria on oysters coated with lacticin NK24 and nisin, when stored at 3 and 10 °C, was found [92]. Results reported a shelf-life extension of 5 to 12 days in the oysters when stored at 10 °C and when treated with bacteriocins (lacticin NK24 and nisin).

Pediocins are thermostable proteinaceous products from *Pediococcus acidilactici* and *Pediococcus pentosaceus*. Pediocins are widely recognized for their antilisterial activity [17,31,93,94]. They are reported to form pores in the cytoplasmic membrane by attaching to the core and forming cracks, causing lysis of the target microorganisms [95].

The antimicrobial activity of the pediocin (bacALP57) isolated from non-fermented shellfish (oysters, mussels, and clams) against non-fermented seafood microorganisms (*Listeria innocua*, *L. monocytogenes*, *Staphylococcus aureus* and *Bacillus cereus*) was observed. The highest inhibitions of *L. monocytogenes* and *Listeria innocua* were found and, generally, inhibitions ranging between 1 and 5 mm were observed for other non-fermented spoilage microbes from seafood [96].

The preservative impacts of pediocin ACCEL (1500 & 3000 IU/mL) from *Pediococcus pentosaceus* on skinless blue shark steak were evaluated by Yin, Wu, & Jiang [97]. The application of the pediocin could significantly reduce the population of aerobic plate counts during storage at 0 and 4 °C in raw fillets. Furthermore, in fish balls, an effective control against *Listeria monocytogenes* [<2 log (CFU/g)] was observed. However, the application of the pediocin at 375 IU/g exhibited a higher inhibition over a higher concentration and control samples.

As per the European Union (EU) directive, nisin was approved for food application from 2011 and widely exploited for application in FFPs [36]. Nisin pre-treated tuna slices packed under vacuum conditions were evaluated for their preservation ability by Sofra, Tsironi, and Taoukis [86]. The inclusion of nisin in osmotic pre-treatment had the lowest microbial growth due to its antimicrobial action.

Bacteriocin GP1 produced from *Lactobacillus rhamnosus* was demonstrated for successfully preserving reef cod [98]. GP1 exhibited the highest inhibition of total viable counts of *Aeromonas* spp., coliforms, *Lactobacillus* spp., and *Vibrio* spp., in comparison to the control, nisin, sodium benzoate and the control sample during storage for 28 days at 0 and 4 °C. Bacteriocin application controlled the spoilage by decreasing the generation of volatile amines (TVB-N and TMA-N) at lower temperature storage conditions. The inclusion of bacteriocins with radiation, vacuum packaging and high-pressure processing has exhibited a synergistic preservation effect and shelf-life extension in seer fish, rainbow trout and dry-cured smoked salmon, respectively [84,99]. A combination of radiation (2 and 5 kGy) and nisin reported a shelf-life extension of 27 and 35 days in seer fish steak during chilled storage. The improvement in microbial quality was attributed to the synergistic effect of irradiation, effective against Gram-negative, and nisin against Gram-positive bacteria [84]. Nisin combined with vacuum packaging inhibited a total viable count [~5 log (CFU/g)], psychrotrophic viable count [~5 log (CFU/g)] and lactic acid bacteria [1.97 log (CFU/g)] in comparison to the control for 16 days of storage [99]. The inclusion of the bacteriocin (EFL4 at 0.64 µg/mL) derived from *Enterococcus faecalis* L04, isolated from *Lateolabrax maculatus* in the form of coating on ready-to-eat salmon fillets, was reported [100]. Bacteriocins exhibited excellent antimicrobial activity against *Staphylococcus putrefaciens* (MIC—0.32 µg/mL) and their contained coating inhibited the growth of the total viable counts. Results also demonstrated the excellent control of muscle degradation (K value) and the generation of amines (TVB-N) responsible for unpleasant sensory attributes. The muscle treated had maintained integrity in comparison to untreated samples due to reduced muscle degradation by the applied bacteriocin. Considering the narrow specificity of bacteriocins, they are used for specific types of spoilage microorganisms. Their highly unstable nature and lower temperature requirement makes them suitable for application in FFPs [38,101].

**Table 1 microorganisms-10-00773-t001:** Preservative effects of the microbial metabolites bacteriocins and organic acids.

Source/Bacteriocin/Organic Acid	Evaluated Matrix	Preservative Effect	Bibliographic References
Bacteriocins			
Combination of glazing with nisin and irradiations treatment at 2 kGy and 5 kGy	Seer fish (*Scomberomorous guttatus*) steaks	Both treatments significantly improved the shelf-life of steaks from 7 days to 32 days (2 kGy) and 42 days (5 kGy) by lowering the microbial and oxidative changes in the steaks.	[84]
Combination of nisin (10 µg/g) with high-pressure processing (450 and 600 MPa) at low temperature (−30 °C)	Dry-cured cold-smoked salmon	The combination treatments significantly inhibited the *Listeria innocua* and other spoilage microbiota of the salmon compared to the control. Additionally, combined treatments improved the sensory properties, peelability and consumer preference of salmon.	[85]
Nisin enriched osmotic solution and vacuum packaging at chilled temperature	Tuna fillets	The combined effect of nisin, osmotic solution and vacuum packaging significantly improved the shelf-life of tuna fillets from 10 days to 51 days at 5 °C by significantly reducing spoilage microorganisms and chemical changes during storage.	[86]
Nisin encapsulated in chitosan microcapsules	Small yellow croaker (*Pseudosciaena polyactis*)	The prepared nisin loaded chitosan microcapsules significantly reduced microbial growth, lipid oxidation, and protein degradation compared to alone hurdle or control, thereby extending shelf-life of croaker by 6–9 days.	[87]
Nisin in combination with tea polyphenols during chilled storage	Tortoise meat (*Trachemys scripta elegans*)	The combined effect of nisin and tea polyphenol treatment to tortoise meat reduced microbial growth, chemical changes and retarded water loss compared to the control.	[88]
Nisin in combination with chitosan. Stored at 4 °C for 8 days	Large yellow croaker (*Pseudosciaena crocea*)	Chitosan (1%) with nisin (0.6%) showed higher efficiency in controlling water loss and other physicochemical indexes, as well as lowered chemical and microbial changes in the fish.	[89]
Nisin-loaded nano-carriers with essential oils (EO) micro-emulsion (EOs used were rosemary, thyme, oregano, and dittany)	Not applied	Enhanced antibacterial activities against *Lactococcuslactis, Staphylococcus aureus, Listeria monocytogens*, and *Bacillus cereus*.	[83]
Biodegradable packaging material (poly lactic acid/sawdust particles) incorporated with bacteriocin 7293	Pangasius fish fillets	Bioactive film effectively inhibited Gram-positive (*Listeria monocytogens* and *S. aureus*) and Gram-negative (*Pseudomonas aeruginosa, Aeromonas hydrphila, Escherichia coli* and *Salmonella thyphimurium*) bacteria.	[90]
Reuterin produced by *Lactobacillus reuteri* INIA P579	Cold smoked salmon	Reuterin effectively inhibited the three different strains of *Listeria monocytogens* in tryptic soy broth assay with concentration range of 2–4 AU/mL. When reuterin was tested on cold-smoked salmon at 8 °C for 15 days and 30 °C for 48 h, *L. monocytogens* counts lowered by 2.0 and 1.0 log (CFU/g) compared to the control.	[102]
Coagulin L1208 from *Bacillus coagulans* L1208	Yellow croaker (*Pseudosciaena crocea*)	Bacteriocin Coagulin L1208 inhibited total viable count, Pseudomonadaceae, *Shewanella*, *Photobacterium* and *Enterobacteriaceae* by producing bacteriostatic ingredients.	[103]
*Enterococcus mundtii* STw38	Fish paste from fresh hake	The bacteriocin inclusion could inhibit native fish spoilage microbiota, especially when packed under vacuum.	[104]
*Streptococcus infantarius*	Trout and tilapia	Inclusion of bacteriocin completely inhibited *Escherichia coli*, *Staphylococcus aureus* and *Listeria monocytogenes* in fish media. While used as wrapping material, it lowered total aerobic count.	[105]
Organic acids			
Aromatic vinegar	Salmon fillets	No impacts on total viable counts and Enterobacteriaceae was found. Superior inhibition of *Pseudomonas* spp. and Psychrotrophic count was observed. Aromatic vinegar had combined effects of phenolics and organic acids in inhibition of microorganisms.	[106]
Citric acid and lactic acid	European hake and megrim	Inhibition of aerobic, anaerobic, psychrotrohic count and Enterobacteriaceae population	[107]
Acetic and ascorbic acid spray	Silver carp (*Hypophthalmichthys molitrix*)	Combination of acetic (1%) and ascorbic (2%) acid exhibited higher inhibition of total viable counts than individual treatments.	[108]
Acetic and citric acid pre-treatment	Bolti Fish (*Tilapia nilotica*)	Combination of acetic acid and citric acid (1 and 3%) exhibited highest inhibition of total viable bacterial count, psychrophilic bacteria, coliform and yeast and mould count. However, the difference was non- significant amongst group and significant in comparison to control.	[109]
Potassium acetate and potassium lactate	Catfish fillet	Combination of organic acid inhibited psychrotrophic bacterial count and extended shelf-life by additional four days.	[110]
Ascorbic, citric and lactic acid based icing	Hake, megrim and angler	Organic acid at 800 mg/kg concentration inhibited mesophilic aerobes in hake, megrim and angler. Additionally, inhibition of psychrophilic and proteolytic bacteria was also found.	[111]
Sodium acetate, sodium lactate or sodium citrate	Salmon	Levels of 2.5% exhibited activity against aerobic and psychrotrophic mircroorganisms, *Pseudomonas* spp., H_2_-S prodcuing, lactic acid and *Entoerobacteriaceae* bacteria.	[112]

### 3.2. Reuterin

Reuterin is a non-proteinaceous low-molecular-weight compound produced by *Lactobacillus reuteri*, exerting the inhibition of microorganisms (Gram-positive and -negative). Reuterin creates imbalances in the microbial cells by the depletion of free thiol groups and the stopping dimer from blocking the enzymatic activity, leading to cell death [113,114]. The aldehyde part of reuterin is also known to inactivate proteins by reacting with the amine group [31,115]. Considering the safety aspects for involving reuterin in a food model, a recent study by Soltani et al. [116] suggested that the maintained cell integrity and improved viability of cells was observed. Furthermore, a better stability of reuterin in the gastrointestinal (GI) tract was found without any toxic effect. Furthermore, the stability of reuterin in combination with other non-thermal techniques has been successfully demonstrated [117]. Reuterin has exhibited antimicrobial activity against the food borne pathogens *E. coli*, *L. monocytogenes*, *S. aureus* and *Salmonella* spp. [118].

Seabass coated with freeze dried *L. reuteri* DSM 26866 (2%) in sodium alginate was evaluated for protective effects (*Angiolillo, Conte, & Del Nobile, 2018*). Reuterin was found in samples fermented with *L. reuteri* for 24 (0.49 g/L) and 48 (0.55 g/L) hours. A significant control of aerobic plate counts, *Pseudomonas* spp., hydrogen sulfite-producing bacteria and Enterobacteriaceae was observed. The inhibitions were attributed to the reuterin produced, extending the microbial shelf-life of the seabass fillets [118]. Additionally, the active coating retained the sensory quality (color and texture) due to the fermentation action. The application of reuterin (10 AU/g), produced by *Lactobacillus reuterin* INIA P579, for the preservation of cold-smoked salmon was evaluated by Montiel et al. [102]. The application of reuterin significantly reduced *Listeria monocytogenes* growth by 2 log units after 15 days of storage. However, reuterin in combination with a high hydrostatic pressure (450 MPa for 5 min.) exhibited effectiveness in the inactivation of *L. monocytogenes* and biogenic amines’ formations in cold-smoked salmon [119]. The combination reduced the microbial population by 3.16 log (CFU/g) in comparison to the control at 4 °C. The authors suggested that the inactivation was due to the synergistic interaction between the high hydrostatic pressure and the natural preservative reuterin in this case. Further, to enhance the bioavailability of reuterin, microencapsulated *Lactobacillus reuteri* with modified atmospheric packaging (95% CO_2_ and 5% O_2_) were evaluated for preserving a tuna burger [120]. Different compositions of sodium alginate had impacts on the generation of reuterin, ranging between 0.640, 0.116 and 0.108 g/L in 2, 1 and 0.5% sodium alginate, respectively. The application of *L. reuterin* reduced the mesophilic bacteria and *Pseudomonas* spp. population at all concentrations evaluated. However, further increased inhibitions were observed in samples packed under modified atmospheric conditions. Combinations of biopreservation methods with other technologies to enhance preservation are suggested.

Reuterin produced and isolated from *Lactobacillus reuteri* INIA P579 was examined against *L. monocytogens* in cold-smoked salmon [119]. The results indicated that reuterin (10 Au/g) significantly lowered *L. monocytogens* counts in cold-smoked salmon by 2.0 and 1.0 log (CFU/g), compared to the control at 8 °C for 15 days and 30 °C for 48 h. Hence, the LAB metabolites enhanced the quality of fish products by lowering the microbial and chemical changes during refrigerated storage. Additionally, the efficacy of the metabolite could be increased by a combination effect with other hurdles or incorporated into emulsion, capsules or active packaging material. The microbial inactivation was found to increase when used in combination with other preservation technology due to a synergistic effect. The inclusion is usually at lower levels and no negative impacts on sensory qualities have been reported.

### 3.3. Bacteriophages and Endolysins

Bacteriophages are specific bacterial viruses, infecting specific groups of bacteria by replicating inside the host, causing lysis of bacteria [121]. Additionally, bacteriophages are known to produce lytic proteins which further extend the antimicrobial activity [122]. Kim et al. [123] reported the effective control of *Vibrio vulnificus* on seafood (abalone) using the novel bacteriophage VVP001. Endolysins are specific antimicrobial peptidoglycan hydrolases synthesized by bacteriophages during multiplication. Due to their safety and non-resistance-development nature, they are evaluated in the preservation of foods [123,124,125,126]. The mechanism of action for endolysins is different for Gram-positive and Gram-negative bacteria, as explained in a recent review by Chang [124].

Combined effects of a bacteriophage cocktail (φ SboM-AG3, φ SsoM-AG8 and φ SboM-AG10) at a concentration of 10^9^ PFU/mL, in combination with a high hydrostatic pressure against *Vibrio cholerae* against salmon fillets and mussels were reported by Ahmadi et al. [127]. The inclusion of bacteriophages reduced *V. cholerae* by 1.2 log (CFU/g) in both salmon and mussels. Additionally, the combination of bacteriophages and a high hydrostatic pressure at 350 MPa further reduced the population by 3.8 and 3.9 log (CFU/g) in salmon and mussels, respectively. The effective decontamination of rainbow trout (*Salmo irideus*) using six bacteriophage cocktails [10^8^ plaque-forming unit (PFU)/mL] was evaluated [128]. Fish samples treated with a bacteriophage cocktail had a shelf-life extension of 3 days. Five bacteriophage cocktails against *Salmonella enterica* in salmon fillets, raw and smoked, were evaluated by Galarce et al. [129]. A lower temperature storage (4 °C) was much more effective in controlling *Salmonella enterica* [130] than a higher temperature (18 °C). The highest inhibition was observed in raw fillets [2.82 log (CFU/g)] in comparison to the smoked [1.16 log (CFU/g)] samples during the 10-day period, reported at 4 °C. The differences in inhibitions were attributed to differences in the water content required for the mobilization of bacteriophages [129].

Pathogenic *Listeria monocytogenes* on tuna sashimi could be reduced by the application of Bacteriophage P100 (5 and 8 log PFU/g). The application of the phage at 8 log PFU/g exhibited promising results by reducing the 4.44 log CFU/g population of *L*. *monocytogenes*. This intervention could increase the safety of ready-to-eat sashimi [131]. The antilisterial activity by Listex P100 was demonstrated on fresh channel fish fillets [132]. A contact time of 15 min was sufficient to reduce the *Listeria monocytogenes* colony population by 1 log CFU/g level on the fish surface. An application dose of 2 × 10^7^ significantly reduced the listerial population and varied at different temperature ranges (4 °C—1.7 to 2.1 log CFU/g, and 10 °C—1.6 to 2.3 log CFU/g). Moreover, the application of Bacteriophage P100 exhibited broad efficiency against Listeria strains from varied strains. Their application in coleslaw food caused a reduction in the contamination of *Listeria monocytogenes* ScottA by 10-fold levels [133]. The bacteriophage from traditional Indonesian ready-to-eat food sources has the ability to inhibit pathogenic *Escherichia coli* on fish meat by 63.78% and 87.89% when incubated for 1 day and 6 days at 4 °C [130]. Recent reports are also available [134] for the inhibition of *Clostridium perfringens* in cooked meats using a bacteriophage obtained from a plant source. The endolysin PlyP825 at a level of 34 µg/mL reduced the inactivation of *L*. *monocytogenes* in smoked salmon (<1 log CFU/g), however, a combination of endolysin with a high hydrostatic pressure (500 MPa) increased this inhibition (>1 log CFU/g) [135]. Bacteriophages and derived endolysins are used for the preservation of foods due to their high host specificity. Considering their lower inhibition of Gram-negative bacteria, they are used in combination with acids or chelating agents, and their ability to inhibit antibiotic resistance, being an anti-biofilm agent, makes them a promising agent for surface decontamination [124].

### 3.4. Organic Acids

Organic acids and their derivatives cause an alteration of pH, the acidification of the cytoplasm impacting the acid-base equilibrium and damaging homeostasis, interfere with gene expression, and hamper cellular metabolism [136,137]. Apart from inhibiting cell activity, organic acids against spore germination have also been reported [38,138]. Davidson, Taylor, and David [38] have suggested the role of the dissociation (proton donation) of organic acids, their ability to generate and transport energy, and inhibit the uptake of nutrients to inhibit the growth of microorganisms. Some recent studies have also reported the inclusion of organic acids in combination due to their synergistic interactions, causing a higher microbial inhibition, which was possibly due to different acid dissociation rates varying the pH [108,139,140]. Organic acids are obtained by different metabolism processes, such as fermentation, oxidation and synthesis [139].

Yadav and Roopesh [14] demonstrated that the spraying of organic acids (lactic or gallic acids) reduced *Salmonella typhimurium* by more than 3.5 log (CFU/cm^2^). Furthermore, a combination of organic acids with a non-thermal technique (atmospheric cold plasma for 30 s) enhanced the microbial inhibition. Organic acids (lactic and gallic acid at 5 to 15 mM) caused the permeabilization of the cell membrane and induced oxidation, causing cell lysis [14]. Citric acid, a commonly used organic preservative, has a reported development of sour taste. Hence, Bou et al. [39] evaluated citric acid in an encapsulated form on ready-to-eat patties from sea bass fillets. The application in an encapsulated form improved their sensory quality. The immersion of cod in citric (5%), lactic (5%) and capric (5%) acid solution inhibited *Pseudomonas* spp., lactic acid bacteria, *Brochothrix thermosphacta*, *Photobacterium* spp. and hydrogen sulfide, producing bacterial counts [140]. The authors’ suggested methods of the application of organic acids have an impact on their antimicrobial activity. Dipping seer fish steaks in sodium-acetate (2%) inhibited the total mesophilic and total psychrotrophic viable counts for 24 days of storage [141]. The author attributed that the lower molecular weight of organic acid inhibited microbial counts, extending the shelf-life by an additional 9 days over the control samples. Recently, SaeidAsr et al. [142] evaluated the effects of a carboxymethyl cellulose coating with essential oils (rosemary) with sodium acetate (2%) on the shelf-life of rainbow trout fillets. The coating containing sodium acetate significantly inhibited the total viable counts, psychrotrophic bacteria, lactic acid bacteria and Enterobacteriaceae over the control and carboxymethyl cellulose. The contradictory growth of *Pseudomonas* spp. increased in the sodium acetated treatment. Positive impacts on the water holding capacity and cooking yield were found to be improved by the sodium-acetate-treated sample due to the reduced pH of the treated samples. Furthermore, the general impacts of microbial metabolites on fishery and fish products are summarized in Figure 2. The application of acetic acid (1%) significantly inhibited the total plate count (>2 Log CFU/g), extending the shelf-life by 3 days. Additionally, the inclusion inhibited lipid oxidation (PV by >25 meq O_2_/kg and TBA by >1.5 mg malondialdehyde/kg) [143]. However, a further combination with chitosan improved the inhibition levels. The inclusion of acetic acid (0.005–0.01%) enhanced the sensory score for pacific white shrimp freshly added and for 12 days of the storage period [28]. Further, their inclusion with other preservatives enhanced microbial (Psychrophilic bacteria, H_2_S-producing bacteria, *Enterobacteriaceae* count) inhibition and quality retention (pH, TVB and TBARS). Organic acids (0.02–5%) have been widely applied for preservation in FFPs and exhibited promising results. Their application method and concentration hampers the sensory quality [144]. The application of organic acids at lower levels, usually below 1%, in combination with other preservative methods are found economical and effective [38].

### 3.5. Other Metabolites

Recently, few studies have been reported on the utilization of LAB as bioprotective cultures or their metabolites (mainly nisin) as biopreservatives for seafood products’ shelf-life extension. In this context, sixteen LAB were isolated from the intestine of *Oreochromis* spp. and investigated for their anti-listeria activity [145]. Among the sixteen isolates, thirteen isolates showed an inhibitory zone on the agar plate inoculated with *Listeria monocytogens*. Furthermore, these microbial isolates were tested in their different forms, including live cells, cell-free supernatant (CFS), alkaline CFS, and heated CFS against *L. monocytogens*. The results indicated that anti-listeria activities occurred by both heat-stable and sensitive compounds, as well as in live cells [145]. Wiernasz et al. [16] investigated six different LAB cultures for the biopreservation of salmon gravlax during 25 days of storage at 8 °C using vacuum packaging. Three of the strains, including *Carnobacterium maltaromaticum* SF1944, *Lactococcus piscium* EU2229, and *Leuconostoc gelidum* EU2249, were competitive in microbial growth, possessing antimicrobial activities against spoilage microorganisms, as well as producing their own metabolic activity. On the other hand, *Vagococcus fluvialis* CD264, *Carnobacterium inhibens* MIP2551 and *Aerococcus viridans* SF1044 were weak competitors, showing weak antimicrobial activities and produced less metabolic activity. However, among all these strains, *C. maltaromaticum* SF1944 showed the highest anti-listeria activity and produced lowered volatilome. In addition, *V. fluvialis* CD264 was capable of preserving the sensory properties and extending the shelf-life beyond 25 days of storage [16]. In their previous study, Wiernasz et al. [69] selected those six LAB strains from 35 different LAB strains, which showed antimicrobial activity, a tolerance to super-chilling and chitosan coating, no antibiotic resistance, and histamine production capacity. Additionally, the biopreservative effects of these six strains were investigated in cod and salmon products alone or in combination with different hurdles, including chitosan coating, super-chilling and modified atmosphere packaging. However, the efficacy of each strain in protecting the quality of cod or salmon was dependent on the type of fish product and the combination of hurdles used [69]. Further, Table 2 details the applications of LAB and metabolites for the preservation of seafood.

Aymerich, Rodríguez, Garriga, and Bover-Cid [146] reported that *Lactobacillus sakei* CTC494 from a meat origin effectively inhibited the spoilage and pathogenic bacteria, and retained the quality of cold-smoked salmon compared to the indigenous LAB strains isolated from same product. However, another study reported that *Lactococcus piscium* EU2241 effectively inhibited the off-odor released by the *Brochothrix thermosphacta* and *Serratia proteamaculans* bacteria in cold-smoked salmon when compared to *Leuconostoc gelidum* EU2247, *Lactobacillus sakei* EU2885, and *Staphylococcus equorum* S030674 [147]. Delcarlo, Parada, Schelegueda, Vallejo, Marguet, and Campos [104] reported that among 132 LAB isolated from mussels of the Argentine coast, only 22 isolates showed anti-bacterial activities against *Listeria innocua* and *L. plantarum*. Interestingly, all 22 isolates belong to the *Enterococcus mundtii* strains, which were confirmed by 16Sr RNA gene phylogenetic analyses. Among the selected isolates, *E. mundtii* Stw38 possesses a higher growth rate and bacteriocin production at 4 °C. When *E. mundtii* Stw38 was applied on fish paste and stored at 4 °C, Stw38 successively survived and lowered the microbiota of fish paste [104]. The combination of the two LAB strains, *Lactobacillus plantarum* AB-1 and *Lactobacillus casei*, was investigated for its synergistic effect against spoilage microorganisms and the quality control of shrimp (*L. vannamei*) during refrigerated storage [148]. The results indicated that the synergistic effect significantly enhanced the antimicrobial activity of *L. plantarum* AB-1 via regulating the AI-2/LuxS quorum sensing system. When shrimps were treated with the co-inoculation of *L. plantarum* AB-1 and *L. casei* and stored for 10 days in the refrigerator, the total volatile basic nitrogen and pH of the samples significantly lowered and the spoilage organism (mainly *Shewanella baltica*) were significantly inhibited [148]. In another study, the combination effect of *Lactococcus piscium* CNCM I-4031 and *Carnobacterium divergens* V41 was investigated for the safety and quality control of peeled and cooked shrimp (*Penaeus vannamei*) during storage at 8 °C for 28 days [149]. The results indicated that there was no synergistic effect of both cultures in controlling the spoilage microorganisms and the co-culture had the same antimicrobial effect as *C. divergens* V41 alone. However, *C. divergens* V41 produced its own metabolic activity which significantly affected the sensory properties of the product. In addition, the *L. piscium* CNCM I-4031 effectively eliminated the activity produced by *C. divergens* V41 in a synergistic effect, thereby maintaining the sensory properties of the shrimp product. Hence, the use of the combination of *L. piscium* CNCM I-4031 and *C.divergens* V41 was recommended for the safety and quality control of shrimp [149]. The multi-bacteriocinogenic *Lactobacillus curvatus* BCS35 culture isolated from the marine origin showed higher antimicrobial activity and stability at 0–2 °C and was used for the biopreservation of fresh young hake (*Merluccius merluccius*) and megrim (*Lepidorhombus boscii*) fish [150]. Additionally, the *L. curvatus* BCS35 culture and cell-free supernatant both significantly lowered spoilage and foodborne pathogenic bacteria, as well as maintained the sensory properties of both fresh fish during refrigerated storage of 14 days [150]. Overall, LAB competes with spoilage or pathogenic organisms for nutrition consumption and renders them dormant. Additionally, LAB secreted metabolites destroy the spoilage or pathogenic microorganisms.

**Table 2 microorganisms-10-00773-t002:** Application of lactic acid bacteria (LAB) and derived metabolites in seafood products’ preservation.

Seafood Products	Lactic Acid Bacteria (LAB)	Effect	Bibliographic References
Not Applied	LAB isolated from intestine of *Oreochromis* sp. Live LAB cells, cell-free supernatant (CFS), alkaline CFS, and heated CFS	Anti-listeria activity, antagonistic activity.	[145]
White leg shrimp	Lactic acid bacterium (*Pediococcus pentosaceus* LJR1)	Inhibited *Staphyloccoccus typhi* (MIC-7.81 µg/mL) and *Listeria monocytogenes* (MIC-15.63 µg/mL) by causing craters and belbs on the microbial surface. Reduction of *L*. *monocytogenes* by 1 log on shrimp was also found.	[151]
Shrimp (*Penaeus vannamei*)	*Lactobacillus plantarum* FGC-12	Significant reduction of total viable count of *Vibrio parahaemolyticus* was observed. Bacteriocin disrupted bacterial cell wall causing lysis.	[152]
Ready-to-eat fish products (sliced surimi and tuna paste)	*Latilactobacillus sakei* CTC494	The microbial inhibition by bacteriocin was dependent upon the product and exhibited antagonistic and mutual interaction on lag phase.	[153]
Sea bass	Mixture of lactic acid bacteria	Antagonistic effect was observed for samples treated with bacteriocin and essential oils. Inhibition of mesophilic aerobic plate count and psychrotrophic bacterial count was observed. Complete inhibition of *Listeria monocytogenes*, coliform, yeast and mold during storage was observed.	[154]
Horse Mackerel fillet	*Lactobacillus plantarum* and *Lactobacillus sakei*	Inclusion of culture reducing *Staphylococcus aureus* by 1 log cycle was observed.	[155]
Mussels	*Lactobacillus plantarum*	Inhibition of *Vibrio* spp. was obtained using bacteriocin.	[156]
Salmon dill gravlax	Spraying of selected LAB cultures and vacuum packaging stored at 8 °C	The strain *Carnobacteriummaltaromaticum* SF1944 had antimicrobial activity against spoilage microbiota and *Listeria monocytogens*. On the other hand, the strain *Vagococcus fluvialis* CD264 had mild antimicrobial activity, but extended the sensory quality of salmon by more than 25 days.	[16]
Three different types of cold-smoked salmon	Spraying method using bacteriocins producing three different strains: *L.curvatus, Carnobacterium maltaromaticum,* and *Lactobacillus sakei* CTC494. Vacuum packaging and storage temperature of 8 °C	*Lactobacillus sakei* CTC494 inhibited the growth of *Listeria monocytogens* and other spoilage microbiota even after 21 days of study, thereby increasing shelf-life of all three types of smoked salmon. However, other two strains limited the pathogens’ growth depending on the type of smoked salmon product. Hence, *Lactobacillus sakei* CTC494 was recommended as a biopreservative for smoked salmon.	[146]
Fish paste	Bacteriocins producing LAB strain (*Enterococcus mundtii*). Vacuum packed and stored at 4 °C	*Enterococcus mundtii* STw38 showed highest activities against Gram-positive bacteria including *Listeria innocua* and *L. plantarum*. Additionally, STw38 strain survived and produce bacteriocins at 4 °C.	[104]
Shrimp (*Litopenaeus vannamei*)	Combined culture of *Lactobacillus plantarum AB-1* and *Lactobacillus casei* applied on the shrimp at refrigerated temperature	The application of combined LAB cultures significantly reduced spoilage microorganisms, mainly *Shewanella baltica*, total volatile base and pH of the shrimp, thereby increasing the shelf-life of the shrimp.	[148]
Peeled and cooked shrimp (*Penaeus vannamei*)	Combination of *Lactococcus piscium* CNCM I-4031 and *C. divergens* V41 applied on the shrimp and packed in modified atmospheric packaging (50% CO_2_ and 50% N_2_) at 8 °C temperature	The results indicated that shrimp treated with combined cultures had higher sensorial properties and lowered microbial and chemical changes at the end of storage time (28 days) compared to the treatment with single LAB culture.	[149]
Cod and salmon based products	Six LAB strains with no histamine production ability, in combination with other hurdles including chitosan, modified atmosphere packaging (MAP), and super chilling.	Improved sensory properties and reduced microbial and chemical changes in cod and salmon products.	[69]
Young hake (*Merluccius merluccius*) and megrim (*Lepidorhombus boscii*)	(1) The multi-bacteriocinogenic *L. curvatus* BCS35 culture; (2) their CFS; (3) Lyophilized bacteriocin powder stored with ice at 0–2 °C for 14 days	The BCS35 culture and their CFS significantly lowered spoilage microorganism as well as *Listeria* spp. Additionally, the sensory properties of both fish were maintained during storage days.	[150]
Cold-smoked salmon	*Lactococcus piscium* EU2241 strain	*Lactococcus piscium* EU2241 prevented the spoilage caused by *Brochothrix thermosphacta* and *Serratia proteamaculans* by acultural and cultural method in cold-smoked salmon, thereby maintaining the sensory properties of the product.	[147]

## 4. Conclusions

In recent years, there has been strong consumer attention for foods preserved with biopreservatives with safe traits. The strong antimicrobial activity of metabolites derived from microorganisms, as well as their safety (no histamine production) characteristics, makes them a promising compound in the preservation of FFPs. FFPs are widely consumed owing to their delicacy and easily digestible protein source. Nevertheless, FFPs are highly susceptible to spoilage during post-harvest and subsequent processes and storage. Therefore, FFPs are often treated with preservatives to inhibit or reduce the growth of microorganisms and control the quality of the products during storage. Biopreservation employs microorganisms or their metabolites to combat undesirable spoilage and pathogenic microorganisms. Biopreservative bacteria, or their metabolites, compete with the undesirable microbiota and dominate the microbiota by utilizing available nutrients. Microbial metabolites (bacteriocins, reuterin, pediocin, lacticin, bacteriophage, organic acids and others) possess antimicrobial activity against a wide spectrum of spoilage and pathogenic microorganisms, including anti-listeria. Numerous studies have confirmed the biopreservative effects of microbial metabolites on different FFPs during different storage and packaging conditions. Additionally, the combination of microbial metabolites with one or more other hurdles, such as a different packaging system (MAP or vacuum packaging) or the addition of another natural antimicrobial (essential oils, etc.) or nano-engineered compounds, greatly enhances the overall antimicrobial activity and thereby increases the shelf-life of the FFPs. Furthermore, the combined effect of various hurdles maintained the quality and sensory properties of the FFPs more than the single hurdle applications. For the inclusion of microbial metabolites as a hurdle technique, it should be combined with two or more hurdles for the high efficiency quality control and shelf-life extension of FFPs. The inclusion of metabolites in the encapsulation process at the appropriate dosage should be established and should explain the impacts of metabolite inclusion in the hurdle concept responsible for the preservation mechanism. The impacts of metabolite inclusion on other spoilage processes related with FFPs shall be addressed. Further, major consideration should be focused on the detailed health risks associated with the employment of microbial metabolites, and the approval from regulation authorities must be focused on.

## Figures and Tables

**Figure 1 microorganisms-10-00773-f001:**
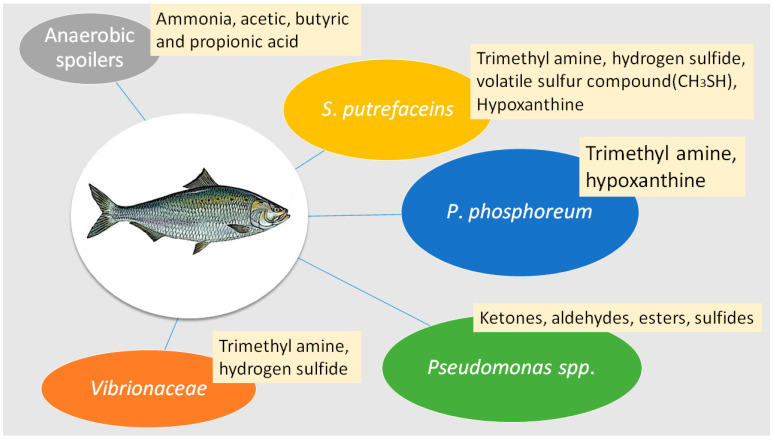
Typical types of spoilage bacteria and typical compounds found during fish spoilage.

**Figure 2 microorganisms-10-00773-f002:**
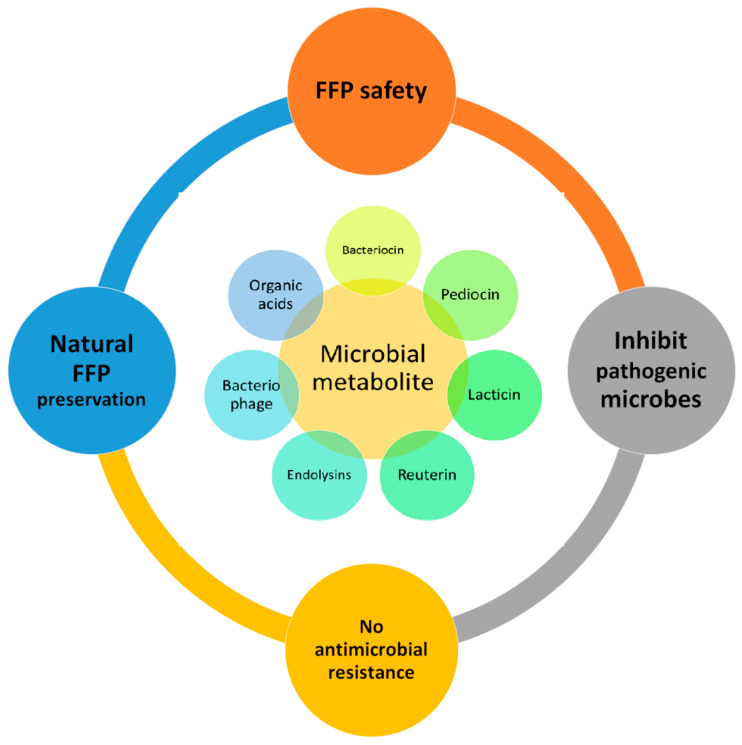
Impacts of microbial metabolites on fishery and fish products (FFP).

## Data Availability

Not applicable.

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
