# Peer review of "Antimicrobial Impacts of Microbial Metabolites on the Preservation of Fish and Fishery Products: A Review with Current Knowledge"

_microorganisms, 2022, doi:10.3390/microorganisms10040773_

Round 1

Reviewer 1 Report

The manuscript includes an interesting review, with wide information on a very interesting concern for seafood consumption. In general terms, I found it correctly designed and presented. However, remarkable aspects ought to be performed before a new revision is carried out.

Abstract

Lines 22-23: Perform.

Lines 21-24 are too general. Contrary, some more information on the real content of the review ought to be provided here.

Line 28: … microorganisms on seafood ?

This section ought to be notably performed.

Introduction

What is the current legal state for the employment of microbial metabolites in seafood and food in general ? This aspect ought to be addressed.

Line 111: Employment of abbreviations ought to be performed. Also in line 115.

Section 2

Figure 1 and line 164: Perform Pseudomonas putrefaciens. In general, revise all Latin names.

Line 177: Inosine and hypoxanthine are mostly the result of endogenous enzyme activity. The authors ought to provide references for supporting the role of bacteria on the formation of such molecules.

Section 3

Line 262: shellfish is mentioned. The Latin name of the species encountered ought to be expressed. The same in line 267. In general, and throughout the whole paper, this information ought to be included.

Line 283: TVB and TMA formation or TVB-N and TMA-N values.

Line 296: Evolution of the K value is considered the result of endogenous enzyme activity. What would be the role of antimicrobials in this case ?

Lines 378-379: A sentence like this ought to be clarified. What were really the experimental conditions ? This criticism ought to be considered in general. Many sentences open questions about the real procedure that was carried out in the reference encountered.

Lines 379-380: The same as for lines 378-379.

Conclusions

Lines 475-488: This is too general and not corresponding to a Conclusions section.

Instead, authors ought to include some more new perspectives of employment of such metabolites. Remarkably, possible health risks and need for approval from institutions ought to be mentioned.

Tables

The authors ought to avoid duplicity of information between Tables and Text.

Author Response

  1. The manuscript includes an interesting review, with wide information on a very interesting concern for seafood consumption. In general terms, I found it correctly designed and presented. However, remarkable aspects ought to be performed before a new revision is carried out.

Response: We would like to thank the reviewer for their valuable suggestion to improve the manuscript.

  1. Lines 21-24 are too general. Contrary, some more information on the real content of the review ought to be provided here.

Response: we have amended the text to meet the reviewer requirement.

  1. Line 28: … microorganisms on seafood?

Response: Said section has been removed 

  1. What is the current legal state for the employment of microbial metabolites in seafood and food in general? This aspect ought to be addressed.

Response: considering the point rightly raised by the reviewer, the regulation has been updated.

  1. Line 111: Employment of abbreviations ought to be performed. Also in line 115.

Response: Said changes have been made to meet uniformity through the text

  1. Figure 1 and line 164: Perform Pseudomonas putrefaciens. In general, revise all Latin names.

Response: Said changes have been made in the text to meet uniformity throught the text. For figure we are suggesting about Shewanella putrefaciens.

  1. Line 177: Inosine and hypoxanthine are mostly the result of endogenous enzyme activity. The authors ought to provide references for supporting the role of bacteria on the formation of such molecules.

Response: we are in agreement with reviewer and have removed the wrongly entered compounds.

  1. Line 262: shellfish is mentioned. The Latin name of the species encountered ought to be expressed. The same in line 267. In general, and throughout the whole paper, this information ought to be included.

Response: We have updated the manuscript with necessary update through the manuscript.

  1. Line 283: TVB and TMA formation or TVB-N and TMA-N values.

Response: Said changes have been made

  1. Line 296: Evolution of the K value is considered the result of endogenous enzyme activity. What would be the role of antimicrobials in this case ?

Response: We wish to clarify; in this case the study has described the role of bacteriocin in quality retention of salmon fillets by reducing ATP degradation. The same has been included in the text.

  1. Lines 378-379: A sentence like this ought to be clarified. What were really the experimental conditions ? This criticism ought to be considered in general. Many sentences open questions about the real procedure that was carried out in the reference encountered.

Response: we have altered the indicated section as suggested by the reviewer to impart clarification

  1. Lines 379-380: The same as for lines 378-379.

Response: we have altered the indicated section as suggested by the reviewer to impart clarification

  1. Conclusions: Lines 475-488: This is too general and not corresponding to a Conclusions section.Instead, authors ought to include some more new perspectives of employment of such metabolites. Remarkably, possible health risks and need for approval from institutions ought to be mentioned.

Response: We are very thankful to the reviewer for their careful observation, and we have modified the conclusion section as suggested.

Reviewer 2 Report

Dear Authors of manuscript entitled „Antimicrobial impacts of microbial metabolites on the preservation of fish and fishery products: A review with current knowledge”, below, please find my comments and suggestions relating with your paper:

  1. In the title, the authors mark "current knowledge" as an important aspect of the presented paper. 130 references were used to write the review article. Among these items I could find very old items, for example possition [35] 1980, [36] 1962, [40] 1977, [41] 1989, [42] 1988, [49] 1961, [50] 1967, [51 ] 1975, [74] 1999, source no 23 is without publication date. The reviewer found 8 publications from 2000-2005, five publications from 2006-2010, 66 publications from 2011-2019, 40 publications from 2020-2021 and one publication from 2022. The reviewer found 8 publications from 2000-2005, five publications from 2006-2010, 66 publications from 2011-2019, 40 publications from 2020-2021 and one publication from 2022. If it is possible, I propose to refer to more scientific articles that were published in the 2020-2022 range.
  2. L 21, In my opinion using abrevages in abstract is not appropriate and necessary. I leave the decision to the authors.
  3. Please ensure that you cite the publications following with Guidelines for Authors, numbers should be put into the [. ].
  4. L44-45, “According to the world health organization (WHO) food-borne illness accounts for around 600 million diseases and 420 thousand deaths per year”, please ensure that this data are true, 600 million diseases sounds like 600 million different type of diseases but probably authors wanted to write about 600 million cases, please rewrite this information.
  5. L47-48, Please add a few examples of synthetic antimicrobial agents that the authors write about.
  6. After reading the text in the range of lines 41-83, I started to wonder what the article is about. The information is provided without thoughtful ordering. Reviewer has the impression that the given text is not logical, the text is written without any order, the information are given randomly and very, very generally.
  7. 114, please, avoid using abbreviations in heads
  8. L 84-113, Again, as before, the text is too general.
  9. Scientific texts require specific numerical details, specific values, and not determining them by using adjectives. This fragment needs to be supplemented with, among others: specific nutritional values, the content of the listed ingredients (fats, vitamins, minerals and others) in% or gL. How long exactly is the shelf life? Writing "relatively shorter" is not enough. What type of enzymes? Please provide their names and the optimal operating temperature. What do these enzymes cause? What kind of microbes is this section about? List their names and describe what spoilage of FFP their cause. List specific examples of specific spoilage organisms, what parasites? SPECIFIC INFORMATION is missing everywhere. I have not found a single example of the specific name of the synthetic preservative FFP, which the authors so often mention in this fragment of the text. The authors write "... several novel non-thermal techniques are widely under evaluation for the preservation of FFP ...", please mention specifically, what techniques are meant.
  10. The authors repeat the same information in several places of the same paragraph.
  11. L123-125, this information is from 1962, please cite a new source, more actual. Similarly with other publications from 1988, 1961 and so on. If the authors cite data, e.g. on microbial contamination of fish, please cite the latest available literature, especially since it is a review article.
  12. Overall, the article contains a lot of information but presented in a very inconsistent way. Too much text is devoted to general descriptions without specific information. The authors describe interesting issues, but please indicate how this review article differs from other available articles on a similar topic?
  13. In my opinion, the article requires an in-depth correction and improvement, and the literature used also needs to be refreshed and rethought.

Author Response

In the title, the authors mark "current knowledge" as an important aspect of the presented paper. 130 references were used to write the review article. Among these items I could find very old items, for example possition [35] 1980, [36] 1962, [40] 1977, [41] 1989, [42] 1988, [49] 1961, [50] 1967, [51 ] 1975, [74] 1999, source no 23 is without publication date. The reviewer found 8 publications from 2000-2005, five publications from 2006-2010, 66 publications from 2011-2019, 40 publications from 2020-2021 and one publication from 2022. The reviewer found 8 publications from 2000-2005, five publications from 2006-2010, 66 publications from 2011-2019, 40 publications from 2020-2021 and one publication from 2022. If it is possible, I propose to refer to more scientific articles that were published in the 2020-2022 range.

Response: We have updated the manuscript with the available recent literature

L 21, In my opinion using abrevages in abstract is not appropriate and necessary. I leave the decision to the authors.

Response: Said changes have been made

Please ensure that you cite the publications following with Guidelines for Authors, numbers should be put into the [. ].

Response: The reference management has been done using the software as recommended by the journal. So, that could be corrected at the later stage.

L44-45, “According to the world health organization (WHO) food-borne illness accounts for around 600 million diseases and 420 thousand deaths per year”, please ensure that this data are true, 600 million diseases sounds like 600 million different type of diseases but probably authors wanted to write about 600 million cases, please rewrite this information.

Response: we are very thankful to the reviewer for the careful observation; we have modified that as infections.

L47-48, Please add a few examples of synthetic antimicrobial agents that the authors write about.

Response: The said details has been updated

After reading the text in the range of lines 41-83, I started to wonder what the article is about. The information is provided without thoughtful ordering. Reviewer has the impression that the given text is not logical, the text is written without any order, the information are given randomly and very, very generally.

Response: We have carefully edited the text and arranged in an orderly manner

114, please, avoid using abbreviations in heads

Response: Said changes have been made

L 84-113, Again, as before, the text is too general.

Response: We have omitted the general text and only kept the introductory important context.

Scientific texts require specific numerical details, specific values, and not determining them by using adjectives.

This fragment needs to be supplemented with, among others: specific nutritional values, the content of the listed ingredients (fats, vitamins, minerals and others) in% or gL.

How long exactly is the shelf life? Writing "relatively shorter" is not enough. What type of enzymes?

Please provide their names and the optimal operating temperature. What do these enzymes cause?

What kind of microbes is this section about? List their names and describe what spoilage of FFP their cause.

List specific examples of specific spoilage organisms, what parasites?

Response: We have updated specific details through the manuscript.

SPECIFIC INFORMATION is missing everywhere.

I have not found a single example of the specific name of the synthetic preservative FFP, which the authors so often mention in this fragment of the text.

Response: Details have been updated.

The authors write "... several novel non-thermal techniques are widely under evaluation for the preservation of FFP ...", please mention specifically, what techniques are meant.

Response: Details have been updated.

The authors repeat the same information in several places of the same paragraph.

Response: We have omitted the repetition

L123-125, this information is from 1962, please cite a new source, more actual. Similarly with other publications from 1988, 1961 and so on. If the authors cite data, e.g. on microbial contamination of fish, please cite the latest available literature, especially since it is a review article.

Response: We have updated the details with recent references

Overall, the article contains a lot of information but presented in a very inconsistent way. Too much text is devoted to general descriptions without specific information. The authors describe interesting issues, but please indicate how this review article differs from other available articles on a similar topic?

Response: This review provides recent developments on preservative action of metabolites derived from microorganisms on seafood. Thus, there could be some overlapping information with the similar topic but this article focuses on the recent advances. Besides, these similarities are inevitable due to discussing the information of the topic.

In my opinion, the article requires an in-depth correction and improvement, and the literature used also needs to be refreshed and rethought.

Response: We have made an in-depth modification to suit the requirement.

Reviewer 3 Report

The biopreservation is a new technique for processing and preservation of foods. This article provides a good reference for the application of biopreservation to fish and fishery products. However, the classification of different biopreservation in the article is recommended to be adjusted. It is also suggested to supplement the safety information of biopreservation as food additives. In addition, the LAB is not the microbial metabolites, it is suggested to adjust the paragraph and narrative of the third part.

#1. In the P2 Line 47-79, it showed that “synthetic antimicrobials used for preservation have been reported to induce resistance, toxicity and health disorders”. But the synthetic antimicrobials used as food additives were all assessed the toxicity to confirm their safety. Please provide more references to verify this statement.

#2 In the P2 Line96 - P3 Line 101, it showed the major problems with synthetic antimicrobials, Please provide more references to verify this statement.

#3 In the P3 Line 101-103, it showed the reason for using naturally derived preservatives. The natural and healthy food are valued by the consumer should also be very important reason. It is suggested to adjust the content of this paragraph.

#4 In the P3 Line 115, it showed that “Fish and fishery products contain a wide array of microorganisms”, please provide more references to verify this statement.

#5 In the P4 Line 140- P5 Line 152, please provide newer references.

#6 In the P5 Line 184 to P13 Line 470, the part of “3.Antimicrobial mechanism of microbial metabolites”, please provide a short summary at the end of paragraph for each antibacterial agent. Including the comparison of the effect of different antibacterial agents or dosage, safety, etc.

#7 Reuterin is also produced by the bacteria (Lactobacillus reuteri), why it was separated into another paragraph. Please explain it or adjust the paragraph of “3.Antimicrobial mechanism of microbial metabolites”

#8 Please provide more references to prove the bacteriophages and endolysins can be safely used as food additives

#9 In the P11 Line 367-400, organic acid can be used as natural antibacterial agent. However, the addition of 5% will cause the decreasing of pH for the fish, which will affect the sensory quality. Please provide more references to prove organic acids have been used to preserve the freshness of practical fish and fishery products.

#10 In the P5 Line 184 to P13 Line 470, the part of “3.Antimicrobial mechanism of microbial metabolites”. The LAB is not the microbial metabolites, and the most of the metabolites described in this article were produced by the LAB. Please adjust the sections in this part.

Author Response

  1. The biopreservation is a new technique for processing and preservation of foods. This article provides a good reference for the application of biopreservation to fish and fishery products. However, the classification of different biopreservation in the article is recommended to be adjusted. It is also suggested to supplement the safety information of biopreservation as food additives. In addition, the LAB is not the microbial metabolites, it is suggested to adjust the paragraph and narrative of the third part.

Response: We would like to thank the reviewer for their valuable suggestion to improve the manuscript. Also we have amended the name of section.

  1. In the P2 Line 47-79, it showed that “synthetic antimicrobials used for preservation have been reported to induce resistance, toxicity and health disorders”. But the synthetic antimicrobials used as food additives were all assessed the toxicity to confirm their safety. Please provide more references to verify this statement.

Response: The details have been updated

  1. In the P2 Line96 - P3 Line 101, it showed the major problems with synthetic antimicrobials, Please provide more references to verify this statement.

Response: Said details have been updated

  1. In the P3 Line 101-103, it showed the reason for using naturally derived preservatives. The natural and healthy food are valued by the consumer should also be very important reason. It is suggested to adjust the content of this paragraph.

Response: We have amended the section as suggested.

  1. In the P3 Line 115, it showed that “Fish and fishery products contain a wide array of microorganisms”, please provide more references to verify this statement.

 Response: Said details have been updated.

  1. In the P4 Line 140- P5 Line 152, please provide newer references.

Response: The details have been updated

  1. In the P5 Line 184 to P13 Line 470, the part of “3.Antimicrobial mechanism of microbial metabolites”, please provide a short summary at the end of paragraph for each antibacterial agent. Including the comparison of the effect of different antibacterial agents or dosage, safety, etc.

Response: The details have been updated at the end of each sections.

  1. Reuterin is also produced by the bacteria (Lactobacillus reuteri), why it was separated into another paragraph. Please explain it or adjust the paragraph of “3.Antimicrobial mechanism of microbial metabolites”

Response: We agree with the reviewer, but due to large availability of literature on reuterin we have tried to specify it in different section.

  1. Please provide more references to prove the bacteriophages and endolysins can be safely used as food additives

Response: We have further extended the section

  1. In the P11 Line 367-400, organic acid can be used as natural antibacterial agent. However, the addition of 5% will cause the decreasing of pH for the fish, which will affect the sensory quality. Please provide more references to prove organic acids have been used to preserve the freshness of practical fish and fishery products.

            Response: We have further extended the section

  1. In the P5 Line 184 to P13 Line 470, the part of “3.Antimicrobial mechanism of microbial metabolites”. The LAB is not the microbial metabolites, and the most of the metabolites described in this article were produced by the LAB. Please adjust the sections in this part.

Response: We have made necessary changes to clarify the confusion.

Round 2

Reviewer 1 Report

The manuscript has been performed according to previous comments. I would recommend acceptation.

Reviewer 2 Report

The manuscript has been greatly revised and improved.